# Recent Advances in Biosensors for Diagnosis of Autoimmune Diseases

**DOI:** 10.3390/s24051510

**Published:** 2024-02-26

**Authors:** Ahlem Teniou, Amina Rhouati, Jean-Louis Marty

**Affiliations:** 1Bioengineering Laboratory, Higher National School of Biotechnology, Constantine 25100, Algeria; a.teniou@ensbiotech.edu.dz (A.T.); a.rhouati@ensbiotech.edu.dz (A.R.); 2Laboratoire BAE, Université de Perpignan through Domitia, 66860 Perpignan, France

**Keywords:** autoimmune disease, biomarkers, biosensors, diagnosis

## Abstract

Over the last decade, autoimmune diseases (ADs) have undergone a significant increase because of genetic and/or environmental factors; therefore, their simple and fast diagnosis is of high importance. The conventional diagnostic techniques for ADs require tedious sample preparation, sophisticated instruments, a dedicated laboratory, and qualified personnel. For these reasons, biosensors could represent a useful alternative to these methods. Biosensors are considered to be promising tools that can be used in clinical analysis for an early diagnosis due to their high sensitivity, simplicity, low cost, possible miniaturization (POCT), and potential ability for real-time analysis. In this review, recently developed biosensors for the detection of autoimmune disease biomarkers are discussed. In the first part, we focus on the main AD biomarkers and the current methods of their detection. Then, we discuss the principles and different types of biosensors. Finally, we overview the characteristics of biosensors based on different bioreceptors reported in the literature.

## 1. Introduction

Autoimmune diseases (ADs) are a group of disorders in which the immune system attacks itself and damages organs, healthy cells, and tissues in the body. Normally, the immune system is designed to recognize targets and foreign substances like bacteria and viruses. However, in autoimmune diseases, the immune system fails to distinguish between self and non-self substances, leading to an attack on healthy tissues, thereby causing inflammation, damage, and dysfunction in the affected organs or systems. Autoimmune diseases can affect various parts of the body, including the skin, joints, muscles, connective tissues, blood vessels, and organs such as the thyroid, pancreas, kidneys, and lungs [1,2,3]. Usually, ADs reflect symptoms that indicate the damage of a specific part of the body [4]. Some common symptoms include the following: fatigue, weight loss, and even cognitive disorders such as depression, which can also be observed for the majority of ADs. The principal trigger of autoimmune diseases is not yet understood, but most studies show that a combination of genetic, environmental, and hormonal factors in addition to lifestyle choices can highly contribute to their development [5,6,7]. ADs affect a significant portion of the population; it is estimated that approximately 5–10% of the global population has been diagnosed, with higher incidences observed for women [8]. Moreover, autoimmune diseases can develop at any age, but many of them tend to have specific age patterns, such as rheumatoid arthritis and lupus, which often manifest between the ages of 15 and 44 [9,10].

Diagnosing autoimmune diseases can be challenging because the symptoms that are non-specific differ from one person to another. Medical professionals may use a combination of medical history, physical examinations, laboratory tests, and imaging studies to make a diagnosis. Principally, an enzyme-linked immunosorbent assay (ELISA), indirect immunofluorescence (IIF), and Western blotting are the most common methods used for the diagnosis of ADs [11,12]. Since these methods are less sensitive, expensive, and time-consuming, extensive studies have been carried out to overcome these limitations and develop novel technology platforms for the diagnosis of autoimmune diseases [11,13]. In this context, biosensors can be used as valuable, promising tools that meet the strong demand for the simple, rapid, accurate, and cost-effective diagnosis of ADs with a better performance compared with that of complex methods of analysis.

Biosensors are analytical devices composed of a biological element, a transducer, and a measurement device. This technology principally relies on a highly selective biological recognition between a bioreceptor and its target, thereby enabling the simple, continuous monitoring of this molecule. Based on the nature of the signal transducer, biosensors can be categorized into various types as follows: optical, electrochemical, thermometric, and piezoelectric systems [14]. In the field of ADs, biosensors can be used for disease diagnosis based on the detection of specific biomarkers using antibodies (Abs), antigens (Ags), and peptides as the recognition elements [15]. Studies have shown that biomarkers have different specificities in autoimmune diseases; some of them are associated with multiple diseases and others are disease-specific. Among these biomarkers, autoantibodies (AAbs) are considered to be the best diagnostic biomarkers that highly participate in the pathogenic process of disease development. Due to their specificity and stability during illness, AAbs can be used to diagnose, monitor, and assess disease severity [16,17]. In addition to AAbs, some chemokines, cytokines, and proteins are also used as potential biomarkers for the diagnosis of autoimmune diseases [18,19].

In this review, we summarize the most common ADs and their specific biomarkers that can be used for the diagnosis and monitoring of these diseases. In addition, various types of biosensors reported in the literature on autoimmune disease detection are also discussed.

## 2. Autoimmune Diseases and Their Biomarkers

Currently, there are more than 80 different types of autoimmune diseases. Here are some common ADs and their specific biomarkers [9,20].

### 2.1. Celiac Disease

Celiac disease (CD) is a chronic autoimmune disorder that affects more than 1% of the population. It is triggered by the consumption of gluten by individuals with a genetic predisposition causing harmful damage to the small intestine [21,22]. In the presence of foods containing gluten, the immune system of the affected patients generates disease-specific antibodies that target both the gluten components (gliadin) and the tissue components (tissue transglutaminase, or tTG). This reaction is accompanied by the production of pro-inflammatory cytokines leading to chronic inflammation of the mucosal and submucosal layers of the small intestine [23,24]. CD patients suffer from several clinical symptoms including chronic diarrhea, weight loss, and abdominal pain. While some patients may present extra-intestinal manifestations (anemia, osteoporosis, fatigue), a significant number of cases remain asymptomatic [25]. The only established and effective treatment for CD is a life-long commitment to a gluten-free diet [21].

Currently, CD diagnosis relies on a biopsy, genetic analysis of human leukocyte antigen (HLA) DQ genes, and serological markers. Studies have shown that the majority of CD patients are HLA-DQ2- and/or HLA-DQ8-positive [26]. While the absence of these genes effectively eliminates CD, their presence alone is insufficient for an accurate diagnosis. For this reason, autoantibodies are also employed for the diagnosis of celiac patients. The main CD-specific antibodies are the transglutaminase antibody (TGA), anti-gliadin antibody (AGA), antibody against deamidated gliadin peptides (DGPA), and endomysial antibody (EMA), and they all belong to immunoglobulin (Ig) classes A and G [27]. These CD antibodies are detectable in the affected mucosa, saliva, and blood, forming the basis of serological detection. Although AGA tests hold historical significance in CD identification, they are no longer routinely recommended due to their comparatively lower sensitivity and specificity in comparison with the TGA and DGPA tests [23,24]. Furthermore, while biopsies of the duodenum or jejunum remain the conventional approach to confirm the diagnosis of CD, the presence of CD-specific autoantibodies may also be used to develop non-invasive tools for CD diagnosis [28,29,30]. However, the combined utilization of clinical observations, laboratory findings, and histopathological evaluations is highly recommended [27].

Recently, REG Ia, which is a molecule associated with tissue regeneration, has emerged as a potential novel biomarker for CD. Its levels are increased in the affected tissue and the blood of celiac patients during periods of damage and inflammation, and they are decreased after following a gluten-free diet [31]. Additional autoantibodies targeting various autoantigens, such as TG-3, actin, ganglioside, collagen, calreticulin, and zonulin, can also be used as biomarkers for intestinal damage in CD patients [32].

Moreover, studies have shown that CD patients present high serum levels of some cytokines such as interleukin (IL)-21, interferon-gamma (IFN-γ), IL-15, IL-4, and IL-10, etc. [33,34]. Some of these molecules remain pertinent for diagnostic purposes as they have distinct specific characteristic alterations in CD that offer insights into the various phases of disease evolution [35].

### 2.2. Multiple Sclerosis

Multiple sclerosis (MS) is a chronic autoimmune disease characterized by inflammation, neurodegeneration, and demyelination of the central nervous system (CNS). It is frequently diagnosed in women aged 20 to 40, although it can affect individuals of all genders, including children and the elderly [36]. MS is principally developed by both genetic predisposition and environmental factors. It induces the destruction of myelin sheaths and myelin-producing oligodendrocytes, thereby leading to the injury and degeneration of demyelinated axons. In these areas where the myelin sheath is damaged, sclerosis (hardened tissues) appear and occur at various positions within the CNS, thus hindering the transmission of electrical signals along the nerves [36,37,38]. This process can be manifests clinically through physical disability, cognitive impairment, and other symptoms that affect quality of life [39].

Clinically, the diagnosis of MS is principally based on a combination of neurological signs and symptoms, magnetic resonance imaging (MRI), the examination of cerebrospinal fluid (CSF) by lumbar puncture, and laboratory tests. MRI is the most frequently used method to validate the diagnosis when distinct lesions are correlated with a specific clinical syndrome; however, certain cases may require supplementary supportive data from CSF analysis and neurophysiological assessments. Moreover, progress in MRI techniques and advancements in serological testing have significantly enhanced the precision in differentiating multiple sclerosis from other disorders [39,40]. In this context, biomarkers play a crucial role in the diagnosis of MS patients as they are easily quantifiable, allowing for the improvement of patient care and disease monitoring [41,42].

The myelin basic protein (MBP) is produced by the oligodendrocytes from the central nervous system and expressed on myelin surfaces; it interacts with the myelin cell membranes and produces adhesion between opposite membranes. It is considered to be one of the main components of the inner layer of the myelin sheath. Since the MBP enters the CSF during demylination, it is used as a potential clinical biomarker for MS diagnosis and treatment monitoring. In addition, multiple studies have shown increased levels of MBPs in the CSF of several MS patients (about 80%). The autoantibodies against the myelin basic protein (anti-MBP) are also suggested as relevant biomarkers since they are present in both the serum and CSF of MS patients [43,44,45].

CNS neurofilaments are intermediate filaments that can be distinguished in heavy, medium, and light neurofilaments as well as α-internexin, and they are released after axonal damage. For MS disease, heavy and light chains are associated with axonal damage. However, recent studies have shown that the light chain (NFL) is a more consistent and reliable biomarker for MS disease [46]. Compared with healthy individuals, MS patients present high levels of the NFL, indicating its importance for MS diagnosis [47,48,49,50,51]. Some studies suggest that the NFL alone or associated with other biomarkers for CNS damage may be considered to be a potential biomarker for neuronal damages [51,52,53].

Oligoclonal bands (OCBs) represent specific patterns of the immunoglobulins observed when analyzing both the blood serum and CSF of patients. These bands are formed by immunoglobulins G (IgG) and M (IgM), which are produced by plasma cells within the CNS [54]. The exclusive presence of these bands in the CSF but not in the serum indicates intrathecal antibody synthesis, thus serving as a significant marker for clinically definitive MS in the majority of patients [55]. It is important to note that OCBs may also be found in other inflammatory CNS diseases [56]. However, for MS patients, the presence of OCBs gives supporting insights into the immune and inflammatory nature of the disease [57,58,59,60].

The IgG index represents the ratio between the IgG levels in the CSF and serum compared with the reference protein albumin [61]. It serves as a marker for the intrathecal production of immunoglobulins, specifically indicating an increased B cell response in the central nervous system, thereby suggesting the presence of MS [62]. Moreover, an increased IgG index is rarely found in MS patients without OCBs. Nevertheless, the IgG index remains a good biomarker for MS diagnosis and is routinely assessed during the diagnostics [60].

In addition to the IgG index, numerous studies have reported increased levels of the glial fibrillary acidic protein (GFAP) in MS patients when compared with those in healthy controls. The GFAP is an astrocyte intermediate filament component that is released as a sign of astrocyte damage and astrogliosis [63,64,65]. Other biomarkers including IL-12, osteopontin (OPN), microRNAs (miRNAs), tau proteins, the neural cell adhesion molecule (NCAM), the nerve growth factor (NGF), and the brain-derived neurotrophic factor (BDNF) are also suggested as complementary biomarkers [45,66,67,68,69,70].

### 2.3. Rheumatoid Arthritis

Rheumatoid arthritis (RA) is a chronic inflammatory disease characterized by inflammation and deformity of the joints, leading to functional limitations, work disability, and an inadequate quality of life. It is one of the most common forms of arthritis; it shows an estimated global prevalence of 0.8% in adults with a higher occurrence among females [71,72]. In general, rheumatologists use a combination of clinical evaluation, imaging studies (like X-rays and MRI), and blood tests to diagnose and monitor the disease. Several biomarkers have been studied and used to identify individuals with RA or those in the pre-clinical stage before symptoms manifest [73,74].

Rheumatoid factors (RFs) are autoantibodies that specifically target the Fc portion of IgG; it is found in the blood of around 70–80% of RA patients [75,76]. For RF antibodies, the IgM-RF isotype is the most frequently used index for diagnosis because of its efficient agglutination [77]. Patients with a positive RF often experience more aggressive disease forms and severe functional impairments. Moreover, RFs are also detected in patients with other autoimmune disorders or infectious diseases (hepatitis C virus, subacute bacterial endocarditis, Epstein–Barr virus). Therefore, RF positivity alone is insufficient for diagnosis [78].

Anti-citrullinated protein antibodies (ACPAs) have also been investigated as potential biomarkers for RA. Citrullination proteins are an abnormal form of protein that occur in inflamed joints and generate novel epitopes that are not tolerated by the immune system, thereby leading to the production of new autoantibodies [79]. These antibodies are capable of predicting joint damage and assessing the development of early RA [80]. Among these ACPAs, the anti-cyclic citrullinated peptide (anti-CCP2) presents an excellent diagnostic impact. Both the rheumatoid factor (RF) and anti-CCP2 show similar sensitivities in diagnosing RA, but the anti-CCP2 exhibits higher specificity [81].

Along with the RF and anti-CCP antibodies, the determination of antibodies against mutated citrullinated vimentin (anti-MCV), which is an antibody in the ACPA family, and the anti-carbamylated protein (anti-CarP) can be useful for RA diagnosis [73,80]. Despite being among the novel biomarkers investigated, these two biomarkers have not exhibited higher sensitivity or specificity when compared with the RF and anti-CCP2 tests, which limit their adaptation in routine clinical practice [80,82].

Numerous studies have demonstrated that the erythrocyte sedimentation rate (ESR) and the C-reactive protein (CRP) measureme.nts remain relevant in the diagnosis of RA [83,84,85]. The ESR indicates the rate at which erythrocytes settle in plasma when suspended in a vertical tube and also serves as a non-specific biomarker for inflammation. Furthermore, with it belonging to the pentraxin protein family, the CRP functions as an acute-phase reactant that is involved in innate immune response [86]. Moreover, RA patients suffer from an excess in pro-inflammatory cytokines that triggers the liver to produce the CRP, making it an attractive disease biomarker [83]. Therefore, the ESR and CRP levels are used to monitor disease activity and response to treatment, but their individual predictive values in RA are not sufficient [87,88].

### 2.4. Psoriatic Arthritis

Psoriatic arthritis (PsA) is a chronic systemic inflammatory disorder that is characterized by immune cell infiltration as well as synovial hyperplasia, and it is often accompanied by active psoriasis or a personal/family history of this affection. PsA is the result of a complex interaction between genetic, immunological, and environmental factors [89]. PsA patients often suffer from pain, swelling, and joint tenderness, thereby leading to limited daily activities and an impaired quality of life [90]. PsA is similar to systemic lupus erythematosus (SLE) and RA. However, research indicates that each disorder originates from an incorporated and interconnected signaling pathway that impacts various components of the immune system. These pathways assume unique functions in the pathogenesis of each disorder [91,92,93]. Moreover, PsA patients face an increased risk of mortality compared with that of the other ADs [94]. The diagnosis of psoriatic arthritis is principally based on physical assessment, the typical absence of the rheumatoid factor, and distinctive radiographic characteristics. Approximately 40% of individuals with psoriatic arthritis experience joint deterioration on radiographs. Hence, an accurate diagnosis and timely intervention can considerably influence disease evolution [95].

At present, there is no specific biomarker used for an accurate PsA diagnosis and treatment response. Therefore, many efforts are needed to help the development of PsA biomarkers and improve diagnosis and patient management [96]. Some studies suggest that the tumor necrosis factor alpha component (TNFα) may be considered to be the principal biomarker for PsA diagnosis since treatment with agents that decrease the levels of the TNF are beneficial for psoriatic arthritis patients [95,97,98]. Moreover, various common biomarkers for other ADs can be used including the ones as follows: the VEGF (vascular endothelial growth factor), P-selectin (a marker for platelet activation), resistin and leptin (adipokines), ACAP, markers for insulin resistance, and indicators of endothelial dysfunction [94,99,100]. In addition, several research findings conclude that individuals with PsA present elevated IL2, IL10, and CRP levels, and they can correlate better with other biomarkers to give insights into disease activity [99,101,102].

### 2.5. Systemic Lupus Erythematosus

Systemic lupus erythematosus is a complex, multifaceted autoimmune disease that is principally characterized by periods of flare and remission (quiescence). It can affect individuals of different ages, especially woman (more than 90% of patients). Genetic predisposition, environmental triggers, and hormonal factors are the principle causes leading to the development of SLE [103]. SLE’s clinical syndrome involves the production of abnormal immunocomplexes and antibodies against cellular components (self-antigens), disturbances in both innate and adaptive immunity, and an imbalance in the production of several cytokines [104]. This disease presents a wide range of symptoms including fatigue, pain, inflammation. and dysfunction of multiple affected organs [105,106].

The diagnosis of SLE is based on a combination of typical clinical symptoms, positive serological tests (ELISA, IF., etc.), and optional biopsies [106]. Various biomarkers have been studied to classify affected individuals, assess disease activity, and determine treatment efficacy. One of the most clinically relevant biomarkers for SLE is that of the presence of antinuclear antibodies (ANAs), particularly those targeting double-stranded DNA (anti-dsDNA) and other nuclear components. Anti-dsDNA is considered to be a highly specific biomarker for SLE diagnosis and is routinely measured in clinical practice to assess disease activity [105,107]. Apart from ANAs, other autoantibodies, such as anti-Smith (anti-Sm), anti-ribonucleoprotein (anti-RNP), the B-cell activation factor (BAFF), a proliferation-induced ligand (APRIL), anti-TRIM21/TROVE2 circulating autoantibodies, and anti-histone antibodies, are also commonly observed in SLE and contribute to the clinical heterogeneity of the disease [108,109].

Cytokines and chemokines are also considered to be critical mediators of the inflammatory response that are involved in the pathogenesis of SLE. Elevated levels of pro-inflammatory cytokines, such as interferon-alpha (IFN-alpha), tumor necrosis factor-alpha (TNF-alpha), IL-6, IL-12, IL-23, IL-1, TGF-β, IL-21, and IL-17, have been detected in SLE patients and are associated with disease activity and organ involvement [104]. Various acute-phase reactants, such as the CRP and ESR, are commonly used as non-specific markers for inflammation in SLE. While these biomarkers lack specificity for SLE, they can provide information about ongoing inflammatory processes [110,111].

### 2.6. Clinical Concentrations of Some AD Biomarkers

Significant advancements in molecular biology and immunology allow for the development of the abovementioned biomarkers for each AD. These biomarkers are found in detectable concentrations in bodily fluids. Table 1 summarizes the clinical concentration levels of some AD biomarkers.

### 2.7. Other Diseases

Sjögren’s syndrome is an autoimmune disease that principally affects the exocrine system of middle-aged women. Its pathophysiology is very close to that of SLE, leading to an increased risk of hypertension and dyslipidemia [119]. Additionally, SS patients experience higher incidences of hypertension, aggravated endothelial dysfunction, increased arterial stiffness, and a higher IMT (intima–media thickness) compared with those observed for healthy individuals [119]. The antiphospholipid antibodies (aPLs) and the lupus coagulant are considered to be the principle SS biomarkers [120]. Furthermore, numerous other autoantibodies such as anti-cardiolipins, IgG, and anti-HDL (considered to be potential biomarkers for SS diagnosis) may have potential implications in Sjögren’s syndrome [121].

Autoimmune hepatitis (AIH) is a rare autoimmune disease that affects the liver, leading to inflammation and liver failure [122]. This type of AD is frequently related to non-specific symptoms including abdominal pain, nausea, fatigue, and arthralgias. Aspartate aminotransferase (AST), alanine aminotransferase (ALT), immunoglobulin G (IgG) levels, the antinuclear antibody (ANA), smooth muscle antibodies (SMAs), anti-liver kidney microsome type 1 (anti-LKM1) antibodies, and soluble liver antigens (SLAs) are considered to be the most used biomarkers for the diagnosis of this type of disease [123].

Crohn’s disease is a chronic inflammatory AD that can affect any part of the gastrointestinal tract leading to ongoing symptoms and complications. This disease is characterized by chronic relapsing as well as the debilitating and remitting inflammation of the gastrointestinal tract [124]. Most Crohn’s patients suffer from some clinical symptoms including abdominal pain and diarrhea while others remain asymptomatic [125]. The most extensively described biomarkers in the literature for the diagnosis of Crohn’s disease are fecal calprotectin (FC), which is a protein released by neutrophils in response to inflammation, and the serum C-reactive protein (CRP) [125]. In addition, a combination of inflammatory, tissue injury. and microbiome-associated biomarkers is also recommended as a complementary biomarker for the diagnosis of Crohn’s disease.

Graves’ disease (GD) is an autoimmune disorder that affects the thyroid gland causing hyperthyroidism. Most Graves’ disease patients suffer from anxiety, weight loss, fatigue, heat intolerance, and palpitations [126]. The most prominent biomarker for Graves’ disease is the presence of autoantibodies targeting the thyroid-stimulating hormone receptor (TSHR). Thyroid-stimulating immunoglobulins (TSIs) and thyrotropin receptor antibodies (TRAbs) are also considered to be autoantibodies associated with Graves’ disease [127]. Additionally, thyroid hormone levels, including elevated levels of free thyroxine (FT4) and triiodothyronine (FT3), coupled with the suppressed thyroid-stimulating hormone (TSH) serve as essential biochemical markers for diagnosing and monitoring Graves’ disease [128].

These are just examples of common autoimmune diseases, and there are many other autoimmune diseases that can affect various organs and systems in the body. Autoimmune diseases can be challenging to diagnose and manage, and treatment often involves a combination of medications to suppress the immune response and manage complications.

## 3. Key Components of Biosensors

In recent decades, biosensors have emerged as potential analytical tools used in different fields such as medicine, agriculture, food safety, and environmental monitoring [129]. A biosensor is a bioanalytical device composed of a specific biorecognition element (enzymes, antibodies, nucleic acids, aptamers, microorganisms, and cells) combined with an adequate transducer (Figure 1) [15]. They aim to provide accurate and reliable real-time information about the analyte. The principle of biosensors is based on a biological recognition that occurs between the bioreceptor and its target. This biological recognition is converted into measurable signals that can be proportional or inversely proportional to the analyte’s concentration [130]. Depending on the nature of the biological recognition process, biosensors can be divided into biocatalytic sensors or affinity biosensors [131]. For biocatalytic sensors, the interaction of the bioreceptor with its specific analyte results in the emergence of a new measurable product. This type of biosensor may use enzymes, microorganisms, tissues, and whole cells as a bioreceptor to catalyze the target. However, this class of biosensor is limited to biological analytes that have corresponding enzymes [132,133]. Therefore, affinity biosensors have emerged as a potential alternative approach taking into consideration interactions between a high variety of bioreceptors with different target molecules. Theses interactions are principally based on the adequate complementarity of the size and shape between the bioreceptor (antibodies, nucleic acids, aptamers) and its target analyte. Furthermore, in this type of biosensor, no chemical product is formed during the formation of the bioreceptor–target complex. Affinity biosensors are characterized by a high sensitivity and selectivity toward the analyte [133]. As autoantibodies are considered to be the principal biomarkers for the diagnosis of autoimmune diseases, the biorecognition mechanism that is most adopted by AD biosensors is one principally based on affinity interactions. The principal transducers used for AD biosensors can be optical or electrochemical [11].

Generally, biosensors can be classified into different categories according to the type of bioreceptor or the type of transducer.

### 3.1. Bioreceptors

The bioreceptor represents the biological part of the biosensor that interacts with the target molecule leading to the production of a measurable signal. It is considered to be the most crucial component since it is responsible for the sensitivity and selectivity of biosensors [134]. The most commonly used bioreceptors are discussed in this section.

Enzymes are natural proteins that serve as biocatalysts; they convert specific substrate molecules into products without being consumed in the reaction. They are widely used as bioreceptors to develop a biosensor with a highly specific binding capability and catalytic activity that are based on the integrity of their native protein conformation [135]. Generally, the working principle of an enzyme-based biosensor depends on the products of the enzyme-catalyzed reactions that can be detected either directly or in conjunction with an indicator [136].

Antibodies (ABs), which are also known as immunoglobulins or glycoproteins, are heavy globular plasma proteins that are considered to be affinity biorecognition elements. They are composed of two heavy and two light polypeptide chains, which form the well-known Y shape. Based on the differences between the heavy chains, the antibodies are classified into five main categories as follows: IgG, IgM, IgA, IgD, and IgE. ABs are produced in animals as an immune response against specific antigens [137,138]. They selectively bind to the target antigen with high affinity and specificity, thereby enabling them to detect the analyte even in the presence of other interfering substances [139].

Nucleic acids are also considered to be bioreceptors. This type of biosensor is principally based on the highly specific affinity binding between two single-stranded DNA chains to form double-stranded DNA generating a measurable signal. The complementarity of the two DNA strands is used extremely to develop rapid and cost-effective tests for infectious diseases, gene analysis, and clinical diagnostics [139].

Molecularly imprinted polymers (MIPs) are synthetic bioreceptors that have the ability to mimic natural recognition elements such as antibodies and other bioreceptors. MIPs are principally formed by creating specific binding sites within the polymer structure [140]. Consequently, the resulting polymer is compatible with the template molecule in terms of size, shape, and functional group position, thus providing high selectivity for the MIP-based biosensors [141].

Aptamers are also synthetic bioreceptors that are selected in vitro from oligonucleotides libraries through the chemical SELEX (systematic evolution of ligands by exponential enrichment) process [142]. They are composed of single-stranded RNA or DNA molecules (10–50 bases) that can bind to a wide range of chemical and biological targets with high affinity and specificity. In the presence of the target, aptamers can fold into 3D structure whose shape and charge complement those of the desired target. The aptamer–target binding is principally based on Van der Waals forces, hydrogen bindings, electrostatic and π-π stacking interactions, and shape complementarity [143,144]. In recent decades, aptamers have emerged as potential chemically stable bioreceptors used to develop different types of biosensors [145].

### 3.2. Transducers

The transducer is the main part of a biosensor that detects the biological signal and converts it to a measurable output. Based on the transducers, a biosensor can be electrochemical, optical, piezoelectric, or thermometric [146,147]. The different types of biosensors are briefly discussed below.

#### 3.2.1. Optical Biosensors

Optical biosensors are composed of a bioreceptor element associated with an optical transducer system. The working principle of the optical transduction processes is based on the changes in the optical properties (phase, absorption, transmission, reflection, amplitude, frequency, etc.) in response to the variations generated by the process of biorecognition. This type of biosensor can be developed using various optical principles, such as colorimetric, fluorescence, chemiluminescence, and surface plasmon resonance strategies [148,149].

#### 3.2.2. Electrochemical Biosensors

Electrochemical (EC) bio-transducers have a wide range of applications in the field of biosensing due to their high sensitivity, specificity, portability, high signal-to-noise ratio, low cost, and fast response [150,151]. The working principle of electrochemical biosensors is based on the variation in electrochemical properties resulting from the reaction that occurs between the bioreceptor and the analyte on the surface of the electrode. This biorecognition interaction produces detectable electrochemical signals in terms of voltage, current, impedance, and capacitance [152]. The assemblage of the target–bioreceptor complex on the electrode surface affects the electron transfer rate, thus providing information about the interaction bindings of the biomolecules [153]. Since most analytes are not electro-active, electrochemical intermediates are used to exchange electrons with the surface of the electrode. Ferrocene is one of the most used intermediates in the electrochemical processes [154]. Based on the principle of transduction, electrochemical biosensors can be classified as follows: potentiometric, impedimetric, conductimetric, and voltammetric. Some of these techniques are privileged because they provide better insight into bimolecular interactions.

These diverse types of electrochemical and optical techniques present different applications, including the detection of pollutants and pathogens as well as the clinical diagnosis of various diseases [155,156].

#### 3.2.3. Piezoelectric Biosensors

Mass change-based bio-transducers are used in many technological applications. This type of bio-transducer is based on a mass-sensitive platform (quartz crystal) that makes the interactions on their surface easily detectable without the application of other reagents. Interactions between the target analyte and the recognition elements create mechanical vibrations that can be translated into a measurable signal [154]. Furthermore, the change in the refractive index due to receptor–ligand interactions is used for analysis [157]. Generally, the piezoelectric effect is exploited to measure strain, force, pressure, or acceleration by converting them into an electrical signal [158].

#### 3.2.4. Thermometric Biosensors

Biosensors using thermal/calorimetric transducers exploit the fundamental characteristics of biological reactions (exothermic or endothermic). This type of biosensor aims to measure the heat resulting from the target–bioreceptor reaction and calibrate it according to the concentration of the target. The change in temperature resulting from biorecognition reactions is determined using a microcalorimeter [159,160]. Calorimetric devices are limited due to relatively long experimental procedures and the lack of specificity in temperature measurement [161].

## 4. Autoimmune Disease Biosensing

Different research groups are working to enhance the diagnostics of autoimmune diseases by developing biosensors and bioassays as promising diagnostic approaches [162,163]. These devices are rapid, sensitive, and cost-effective with an improved analytical performance. According to the literature, most AD biosensors are mainly based on electrochemical detection, but some optical and piezoelectric biosensors have also been reported (Appendix A) [164]. The following overview presents the most recently reported AD biosensors.

### 4.1. Celiac Disease Sensing Strategies

Celiac disease patients suffer from the difficulty of diagnosis because of its multifaceted presentations. The diagnosis of CD is mainly based on serologic testing for specific serum antibodies [165]. The detection of immunoglobulin A anti-tissue transglutaminase antibodies (IgA-TGAs) is the most sensitive and specific test for CD diagnosis. The assessment of IgG DGPA levels is also a frequently used biomarker for CD diagnosis. In addition, IgG DGPAs and IgG TGAs are recommended for children younger than 2 years of age with negative IgA-TGAs and suspected CD [166].

Bonham and his colleagues developed a biosensor using electrochemical DNA (E-DNA) for the detection of IgA TGA antibodies circulating in the blood. In this approach, they hybridized an incorporated DNA oligonucleotide through thiol bonding to the gold electrode surface with a complementary peptide nucleic acid (PNA) chimera, which was specifically a PNA oligomer covalently bound to the gliadin-tTG synthetic neoepitope at its distal end. The incorporated neoepitope was used to enable the binding of the IgA TGA CD-specific AABs, causing a conformational change in the biosensor as well as a change in the environment of an attached redox reporter (methylene blue), thereby producing a measurable current reduction. The developed biosensor showed a limit of detection (LOD) of about 10^−2^ U/mL with a linear range from 0.01 U/mL to 10 U/mL. Despite the high sensitivity and specificity obtained for the detection of the TGA, no measurable response was detected when the assessments were performed on an undiluted whole-blood serum. In addition, this platform presented a relatively high error during measurements, which limits their utilization. Based on this, we can say that more efforts should be aimed at significantly reducing the observed error and increasing the signal quality, thereby enabling the detection in serum [167].

Several sandwich-type sensors using horseradish peroxidase (HRP)-labeled Ig secondary antibodies were developed [168,169,170,171]. In this context, Wajs et al. developed a supramolecular biosensor based on the formation of a sandwich immunocomplex between the gliadin units immobilized on the synthesized polymer, the target antibody, and a reporter anti-mouse–HRP conjugate. For this, polypyrrole–cyclodextrin-modified electrodes were used as a support for the immobilization of a biofunctionalized polysaccharide carrying adamantane units and gliadin units. A screen-printed gold electrode was used as the support for the electropolymerizing of the pyrrole–carboxylic acid films followed by the covalent attachment of te cyclodextrin units and then attaching gliadin by *N*-(3-dimethylaminopropyl)-*N*′-ethylcarbodiimide/*N*-hydroxysulfosuccinimide (EDC/NHS) coupling. After incubation with the target, the modified electrode was further incubated with a HRP-labeled antibody. The amperometric measurements were linear with an anti-gliadin antibody concentration range of 0–10 μg/mL and a limit of detection of 33 ng/mL. The obtained results indicated that the combination of the polypyrrole film with the cyclodextrin/adamantane interface plays a crucial role in enhancing the sensitivity of the biosensor [172].

An interesting concept for the point-of-care diagnostics of CD was reported by Gianneto et al. The group developed a portable device exploiting the integration of screen-printed, electrode-based immunosensors and the remote-controlled Internet of Things-wireless Fidelity (IoT-WiFi) acquisition board. The immunodevice was based on the chemisorption of the open-tissue transglutaminase enzyme on the surface of the gold nanoparticle-functionalized carbon screen-printed electrodes. The IgA and IgG anti-tissue transglutaminase antibodies were captured by the immobilized bioreceptor as highly specific biomarkers related to CD (Figure 2A). When the amperometric signal was generated through a secondary Ab labeled with alkaline phosphatase, it was processed through an on-purpose developed IoT-WiFi-integrated board, allowing for real-time data sharing with doctors. The obtained results showed a diagnostically useful LOD of 3.2 AU/mL (IgA) and 1.4 AU/mL (IgG). The developed device is simple, portable, user-friendly, and presents high versatility and transversality. However, it is not intended for a quantitative assay; it is designed as a screening tool usable for out-of-hospital use and self-diagnosis to efficiently discriminate between anti-TGA IgA/IgG-positive and anti-TGA IgA/IgG-negative serum samples [173].

Improved analytical performances were demonstrated for immunosensors based on nanoelectrode ensembles (NEEs). In this context, novel electrochemical immunosensors based on the use of NEE technology were developed for the detection of IgG-TGA by Ugo and his colleagues [112]. To this end, they functionalized the polycarbonate (PC) membrane-templated gold NEE with tissue transglutaminase. After the target recognition, the NEE was incubated with a secondary antibody labeled with horseradish peroxidase in the presence of H_2_O_2_ as a substrate and hydroquinone as a redox mediator to generate the signal (Figure 2B). In the presence of the target, the generated electrocatalytic current was proportional to the amount of TGA captured from the sample. The optimized system allowed for a detection limit of 1.8 ng/mL, with satisfactory selectivity and reproducibility. However, cathodic detection using HRP and hydroquinone (HQ) not only decreases the stability of the assay but also takes more time and provides a low signal/noise ratio. To overcome these limitations, another study was conducted by Ugo’s team, where they exploited the advantages of NEE technology to develop a novel NEE biosensor for the detection of IgA-TGA. For this, tTG was first immobilized on the polycarbonate membranes of the NEE and used to capture the target TGA. A secondary anti-IgA antibody (binding specifically to the IgA isotype) was labeled this time with glucose oxidase (Gox) and used to generate a signal using the glucose as an electrolyte. This secondary antibody was also functionalized by (ferrocenylmethyl)trimethylammonium (FA^+^) as a redox mediator, and the electrocatalytic signal was produced after the application of an anodic potential (Figure 2C). The developed immunosensor provided a low detection limit of 0.7 U/mL with a linear range between 0.25 and 8.54 U/mL. The proposed strategy could be used for the quantification of IgA-TGA in serum samples from celiac patients. It provides a satisfactory agreement with the outcome obtained by the “classical” fluorenzyme immunoassay method [174]. In comparison with the previous study, the use of GOx make the assay more stable. Furthermore, despite the fact that FA^+^ is an efficient mediator for this GOx, it undergoes highly reversible electrochemical oxidation at moderately positive potential values with fast heterogeneous electron transfer kinetics, thereby providing the best signal/noise ratios achievable with NEEs. Future efforts should be focused on advanced analytical strategies to separately detect the isotypes of the TGA, allowing for differentiation between the voltammetric signals generated by the different isotypes. The possibility of detecting separated signals for IgA and IgG with the same platform looks interesting for improving the early diagnosis of celiac disease, especially for children and IgA-deficient patients.

Nanoelectrode ensemble technology has been also successfully used in electrochemiluminescence (ECL) detection methods. In this context, a novel ruthenium (Ru)-based electrochemiluminescence immunosensor was designed for the TGA assay for CD diagnosis using membrane-templated gold NEEs as a detection platform. For this, tTG was first immobilized as a capturing agent on the PC surface of the track-etched templating membrane. When it captured the target (TGA), it allowed for the immobilization of a streptavidin-modified ruthenium-based ECL label through a reaction with a suitable biotinylated secondary antibody. The application of an oxidizing potential in a tri-n-propylamine (TPrA) solution generated a significant ECL signal. The authors reported a linear range between 1.5 ng/mL and 10 μg/mL and a LOD of 0.5 ng/mL [176]. The main advantage of this system is related to the ECL emission, which is not initiated by the electrochemical oxidation of Ru(bpy)_3_ ^2+^. Instead, it is obtained by applying a potential of 0.88 V versus Ag/AgCl. This lower operative potential significantly reduces the possible interference in samples containing oxidizable species and minimizes possible electrochemical damage to sensitive biomolecules.

Neves et al. developed the first electrochemical immunosensor to assess the DGPA in real serum samples. The developed platform was based on the modification of a disposable screen-printed carbon electrode with carboxylated multi-walled carbon nanotubes (MWCNTs) followed by the electrochemical deposition of gold nanoparticles. After that, a fusion protein of four deamidated gliadin peptides (DGPx4) was deposited. Anti-human IgG labeled with alkaline phosphatase (AP) was used as a secondary antibody, and the sensor response was based on the enzymatic deposition of metallic silver catalyzed by AP. The developed sensor provided comparable results and was more cost-effective than the commercial reference ELISA kit. However, in future studies, it would be interesting to use dual screen-printed electrodes to endorse the simultaneous detection of antibodies against DPG (IgG) and tissue transglutaminase (IgA) in order to facilitate a diagnostic work-up and the pursue of celiac disease [177].

The combination of two or more tests is highly recommended as it provides better results with a high sensitivity and a proper diagnosis. For this, CosaGarcia’s group developed the first electrochemical dual immunosensor for the simultaneous detection of IgA- and IgG-type AGA and TGA antibodies in samples of real patients. The proposed system was based on a dual screen-printed carbon electrode, with two working electrodes that were nanostructured with a carbon–metal hybrid system that works as the transducer surface. For this, the nanostructured electrode surface was functionalized with gliadin and tTG. In the presence of the targets, the electrochemical measurements were recorded using AP-labeled anti-human antibodies and a mixture of 3-indoxyl phosphate with silver ions that acts as a substrate (Figure 2D). The obtained results showed a LOD of 2.45 U/mL for TGA IgA detection and 2.95 U/mL for TGA IgG detection. Furthermore, the LOD was 3.16 U/mL for AGA IgA detection and 2.82 U/mL for AGA IgG determination. The obtained results were corroborated using commercial ELISA kits, thereby indicating that both methodologies are in good agreement. The developed biosensor presents an interesting analytical performance, and it can be a good alternative to the traditional methods offering exciting opportunities through point-of-care strategies. Future efforts should be focused on reducing and simplifying the fabrication steps of this platform [175].

In order to provide cost-effective sensing strategies, label-free sensors have been developed instead of sandwich-type sensors that are labeled. For this, Gupta et al. [178] fabricated a sensitive label-free immunosensor using graphene quantum dots (GQDs)/polyamidoamine (PAMAM) nanohybrids modified on gold nanoparticles (AuNPs) and embedded in MWCNT. For this, the working electrode (MWCNT-AuNP) was functionalized with cysteine, which acts as a linker between Au and GQDs, and the amino groups were further coupled with carboxyl of the graphene quantum dots (GQDs) through a carbodiimide bond. After this, several tTG antigenic probes were covalently linked with PAMAM using EDC/NHS chemistry. This sensor presented ultrahigh sensitivity with a limit of detection of about 20 fg/mL. These results indicate the importance of label-free detection and the immobilization strategy, which prevent the loss of the antigenic properties of tTG and the effect of AuNPs and GQDs in enhancing the platform’s sensitivity.

### 4.2. Multiple Sclerosis Sensing Strategies

MS is the most common progressive inflammatory autoimmune disease that causes a severe effect on the central nervous system. The number of patients diagnosed with MS is increasing, yet the diagnostic process of the disease remains complex. The application of biosensors in detecting MS biomarkers offers several advantages including sensitivity, rapidity, and cost-effectiveness [40].

The first immunosensor for MS diagnosis was published in 2007 by La Belle and her colleagues who developed a label-free electrochemical impedance spectroscopy biosensor for the detection of IL-12 in serum. For this, anti-IL-12 IgG was immobilized on the surface of a gold-coated silver ribbon electrode with 16-mercaptohecadecanoic acid (16-MHDA) and an EDC/NHS crosslinking reaction. The developed immunosensor measured the concentration of interleukin-12 in the range of 0–100 pg/mL. However, nonlinearity was found at around 5.0 pg/mL, thereby demonstrating that the detection of the target at physiological levels was between the limited ranges. With further improvements and optimizations, the obtained nonlinearity could be enhanced, allowing for a much higher dynamic range [179]. Two years later, almost the same team of researchers published their next study on this topic. They reported the first label-free EIS-based biosensor for IL-12 detection using a disposable printed circuit board (PCB) electrode system. For this, they electrodeposited gold onto a PCB electrode, but this time they further immobilized the anti-IL-12 monoclonal antibody (MAb) concentration onto the electrode surface through 16-MHDA and EDC/NHS crosslinking. The disposable sensor showed a higher linear range between 0.1 and 500 pg/mL, and a LOD of 3.5 pg/mL was achieved. The analytical performance was examined in complex matrices to test the non-specific reactions. The obtained results demonstrate that this immunosensor is robust and highly sensitive using a simple fabrication technique [180].

In another study, Derkus et al. also designed a simple label-free electrochemical immunosensing platform for anti-MBP detection. For this, they chemically immobilized the MBP on an alginate and alginate-titanium dioxide (TiO_2_) nanocomposite-film-modified platinum electrode (Figure 3A). The results showed that the properties of alginate in the presence of TiO_2_ were improved by enhancing the rate of the electron transfer. The EIS measurements of the alginate-TiO_2_ nanocomposite revealed a lower detection limit of 0.18 ng/mL in comparison with that of the LOD obtained using an alginate MBP (0.25 ng/mL). Furthermore, the developed impedimetric biosensor presented a short response time (104 s for the alginate-MBP immunosensor and 79 s for the alginate-TiO_2_-MBP immunosensor) in comparison with that through the ELISA method. In addition, it was shown that this system can directly determine the anti-MBP levels in blood samples without the need of CSF [181]. The developed system is considered to be a relevant alternative for the determination of anti-MBPs since it is simple, practical, highly sensitive, and cost-effective.

Field-effect transistor technology has also been used to develop biosensors for MS diagnosis. In this context, Guerrero et al. developed a new device based on enzyme field-effect transistor (EnFET) technology for the diagnosis of demyelinating diseases. In this work, the gate of an ion-sensitive field-effect transistor (ISFET) was modified with a synthetic polymer to entrap the desired analyte that contains the MBP. Taking into consideration the MBP chemical structure, a hybrid polymer gel was used for the analyte entrapment as a synthetic receptor, which was formed by polyvinyl alcohol (PVA), tetraethyl orthosilicate (TEOS), and glutaraldehyde (GA). The developed device showed a detection range of 0.1–100 nM [184]. Three years later, Jian Song et al. studied the different bioreceptor layers for MBP detection to determine which polymer matrix provides the best antibody immobilization and sensitivity using fluorescence intensity measurements. Four polymers were tested as follows: polystyrene-co-methacrylic acid (PS-MA), polystyrene-block-poly (acrylic acid) (PS-PAA), Poly (methyl methacrylate-co-methacrylic acid) (PMMA-MA), and Poly (D, L-lactide-block-acrylic acid) (PDLLA-PAA). After that, the best polymers selected were used to develop an organic FET. The results showed that among the tested receptor layers, the PS-MA polymer showed the best performance for anti-MBP immobilization as observed by fluorescence intensity. It was used as a bioreceptor for the construction of an organic field-effect transistor (OFET) biosensing platform showing a linear range between 1 and 500 ng/mL with an optimal sensitivity of 11.9 ± 3.8% for 100 ng/mL of the MBP solution and 1.5 ± 0.3% for 1 ng/mL, which was taken as the detection limit [185].

In another study, Guerrero et al. developed an amperometric biosensor for the determination of anti-MBPs [182]. The developed system was based on carboxylated magnetic microparticles (cMBs) that were used as a solid support for the covalent attachment of MBPs followed by the specific capture of the target antibodies against this protein. After detection, the immobilized complex was further incubated with a secondary antibody labeled with HRP (HRP-anti-hIgG). The resulting complex of HRP-anti-hIgG-anti-MBP-MBP-MBs was magnetically captured on the surface of a screen-printed carbon electrode (SPCE), and amperometric transduction was performed by adding hydrogen peroxide and using HQ as a redox mediator (Figure 3B). The developed bio-platform exhibited a linear range between 0.05 and 50 ng/mL with an achieved LOD value of 0.016 ng/mL. This system was successfully applied to the analysis of serum samples of healthy individuals and patients diagnosed with MS, providing results in agreement with the ELISA methodology. The obtained analytical performance is challenging in terms of sensitivity and the range of linearity in comparison with the literature as well as with commercially available ELISA kits, showing a remarkably shorter assay time (80 min). The preparation of the immunocomplexes in a homogeneous solution through a single incubation step allows for the avoidance of a need of electrode surface modification by applying several steps to achieve the efficient and stable incorporation of the recognition element, which makes this simple and cost-effective. In addition, the use of MBs as an immobilization support enhances biomarker isolation and preconcentration from the clinical samples, thus minimizing unspecific adsorptions. In order to avoid the use of a labeled secondary antibody, aptamers are used as an alternative. In this context, Frank and his colleagues developed a bioluminescent solid-phase sandwich-type micro-assay to detect MS-associated autoantibodies in human serums. The assay was based on two different 20-fluoro-pyrimidine (20-F-Py)-modified RNA aptamers against the anti-MBP autoantibodies as bio-specific elements and the use of obelin as a highly sensitive bioluminescent reporter. The surface microplate wells were first activated with streptavidin and then incubated with Apt12-2b-biotine. After incubation with the target, the conjugate Apt2-9c-obelin was added. The bioluminescence measurements demonstrated a sensitivity of 63.7% and a specificity of 94.2%. The developed micro-assay was applied to analyze the serum samples of healthy donors and MS patients, showing significant results [186].

Since tau proteins are involved in many neurological disorders, they can be used as a complementary biomarker for MS diagnosis. Moreover, several findings have shown that higher levels of tau proteins in CSF are strongly related with MS disease [187]. In this context, Derkus et al. designed an electrochemical nano-immunosensor for the simultaneous quantification of MBPs and tau proteins in the CSF and serum of MS patients. For this, the SPCE surface was functionalized with graphene oxide (GO) nanoparticles and amine-functionalized tris [poly (propyleneglycol)] (pPG) dendrimers. After that, the synthesized GO/pPG nanocomposite structure was used for the immobilization of antibodies of the MBPs and tau proteins. A secondary-antibody-conjugated carboxyl functionalized with pPG/cadmium sulfide (CdS) and pPG/lead sulfide (PbS) probes was used to obtain a sandwich complex. The developed system was characterized and optimized by evaluating the Cd^2+^ and Pb^2+^ electrochemical signals obtained by the ionization effect of the nitric acid. The developed nano-immunosensor showed a linear range between 58 and 227 nM for the MBPs, whereas the tau protein levels were within the 0.5–15.1 nM range, with detection limits of 0.30 nM for the MBPs and 0.15 nM for the tau proteins, which were sufficient for the levels to be analyzed in the neuro-clinic. This platform was also tested using the CSF and serum of MS patients and it showed that the designed nano-immunosensor was capable of detecting proteins with high sensitivity and selectivity. The obtained results presented good agreement with the commercially available ELISA. However, EIS characterization showed nonlinear behavior for the final structures; therefore, more consideration should be given to the immobilization strategy of the antibodies [188].

Different biosensors have been developed for the sensitive detection of osteopontin, BDNF, and mi-RNA [189]. In order to avoid the complicated steps involved in the electrochemical biosensors, Mukama et al. developed a lateral flow aptasensor for OPN detection. For this, they used a biotinylated DNA aptamer for capturing OPN from samples, and an antibody for OPN was immobilized on the test line for the identification of a second specific target, and streptavidin-modified gold nanoparticles were sprayed on the conjugation pad for color detection (Figure 3C). This sensing assay was applied in a concentration range of 10–500 ng/mL and showed a LOD of 10.3 ng/mL. The experiments showed satisfying results in the presence of other different proteins in the PBS buffer and in human serum, showing high specificity and selectivity [183]. Later in 2022, Omidinia and his colleagues designed a label-free electrochemical aptasensor for miR-155 detection. A nanocomposite of SWCNT and polypyrrole (Ppy) was deposited on a graphite sheet substrate and modified with an aptamer as the miR-155 capture probe. The sensing system exhibited a dynamic range from 10 aM to 1 µM with a LOD of 10 aM. Furthermore, the applicability was successfully tested in human serum with good selectivity to differentiate between the complementary target and the non-complementary one [190]. Recently, Akinrinade George Ayankojo et al. developed an ultrasensitive electrochemical biosensor using MIPs integrated into a system of thin-film metal electrodes for the rapid detection of the BDNF as a potential biomarker for neurodegenerative disorders. The performance of the system was studied by the differential-pulse voltammetric technique. The sensor demonstrated a linear range between 10 and 40 pg/mL and a detection limit of 9 pg/mL for BDNF, displaying good selectivity in the presence of closely related neurotrophic factor proteins (CDNF and MANF) as well as for proteins with comparable pI values (such as CD48). The sensor’s selectivity was further validated through competitive and non-competitive binding assays, where the BDNF was found to inhibit the binding of interfering proteins. The applicability of the developed platform was also evaluated using fetal bovine serum. The presented robust sensing platform showed strong affinity, specificity, and cost-effectiveness in diagnosing neurodegenerative disorders [191].

### 4.3. Rheumatoid Arthritis Sensing Strategies

RA is a chronic AD that principally causes joint pain and deformation leading to impaired physical function disability. The rapid diagnosis of RA is essential for better treatment and monitoring of the disease. In this context, several biosensors have been developed for RA diagnosis. RF and anti-CCP are considered to be common biomarkers found in the serum of patients diagnosed with RA. In addition, the tumor necrosis factor (TNF_), CRP, IL-6, and osteopontin are also considered to be non-specific biomarkers for RA diagnosis [192].

In 2018, Zhao et al. developed a label-free ECL immunosensor based on asymmetric heterogeneous polyaniline-gold (PANI-Au) nanomaterial for the quantification of anti-CCP. For this, PANI-Au nanomaterials were synthesized using a simple one-step interfacial method and incorporated with graphite-like carbon nitride (g-C_3_N_4_). The incorporation of PANI-Au not only increased the interfacial stability but also enhanced the ECL efficiency of g-C_3_N_4_. In addition, the use of Au facilitated the immobilization of CCP, which is the recognition element for the anti-CCP antibody, on the electrode surface. The developed sensing platform showed a wide linear range of 0.001–15 ng/mL with a LOD of 0.2 pg/mL. It shows high selectivity, reproducibility, and stability, thereby indicating its application for real samples [193]. In another study, Selvam et al. used a molybdenum disulfide (MoS_2_) and PANI-modified SPE to develop an electrochemical immunosensor for anti-CCP detection. For this, the SPE was modified by drop-casting a MoS_2_ ink to fabricate the electrode base matrix; then a PANI fine layer was electrodeposited on MoS_2_/SPE. After that, the CCP, which acts as a bioreceptor, was immobilized on PANI/MoS_2_/SPE using EDC-NHS crosslinking chemistry (Figure 4A). A PANI-Au nano-matrix was also used to entrap anti-CCP antibodies with higher capacity through physical adsorption and the electrostatic interaction between the AuNPs and anti-CCP antibodies, thereby resulting in a more sensitive immunosensor. The square wave voltammetric measurements showed good linearity between 0.25 and 1500.0 IU/mL of anti-CCP with limit of detection values of 0.16 IU/mL for PBS and 0.22 IU/mL for human serum. The sensor showed high selectivity and stability with the possibility of being applied for real-time anti-CCP detection [194]. Because of its durability, conductivity, and high capacity to incorporate nanoparticles, PANI has been used to develop both electrochemical sensing platforms, resulting in higher sensitivity and better electrode behavior.

In 2019, Veigas et al. developed a single-step colorimetric immunosensor for IgM-RF quantification. The colorimetric immunosensor was based on gold nanoparticles, to which IgG-Fc was covalently linked through a bi-functional polyethylene glycol derivative. In the presence of the IgM RF, which acts as a crosslinking agent, extensive aggregation of the functionalized gold nanoparticles occurred, thereby resulting in a color change from red to purple that can be easily observed by the naked eye. The developed system showed a LOD of 4.15 U/mL of the IgM RF. In comparison with other detection platforms, the proposed sensing strategy did not require any secondary antibodies, additional washing steps, or signal amplification protocols, thus making it simple, rapid, and cost-effective [196].

Later on, different technologies based on the integration and miniaturization of electrodes were used to develop electrochemical biosensors for RA diagnosis. In 2019, Chinnadayyala et al. reported the label-free detection of the IgM-RF based on an interfingered wave microelectrode array (IDWμE). The surface of the electrode was functionalized with a thioctic acid self-assembled monolayer (SAM) for the covalent immobilization of IgG-Fc (Figure 4B). The impedimetric measurements presented a linear concentration range between 1 and 200 IU/mL with a LOD of 0.22 IU/mL. In addition, this technique enabled the quantification of the IgM-RF without the use of labels, which makes it simple and cost-effective [195]. One year later, Chinnadayyala et al. also exploited the same technology proposed before (IDWμE) to develop the first report on a miniaturized interdigitated chain-shaped microelectrode array (ICE) configuration that was used this time for anti-CCP impedimetric detection. For this, the fabricated ICE was functionalized with a SAM-modified Mercaptohexanoic acid (MHA). Then, the synthetic biotin-conjugated cyclic-citrullinated peptide (B-CCP) was immobilized onto the surface of the SAM-modified ICE through an avidin–biotin interaction system. The immunosensor displayed a good dynamic range between 1 and 800 IU/mL with a limit of detection of 0.60 IU/mL and 0.82 IU/mL for PBS and human serum, respectively. Both of the developed sensing platforma showed good reproducibility, selectivity, and stability, thereby indicating their potential application for human serum samples. In addition, their microscale design may facilitate the assembly of the ICE and IDWµE with a portable, miniaturized potentiostat, making them suitable for POCT [118]. However, the use of the avidin–streptavidin immobilization system makes the developed platform more expensive. To overcome this problem, Zhou et al., developed an electrochemical immunosensor using IDE technology for the voltammetric sensing of the anti-CCP. But this time, iron oxide nanoparticles (IONPs) synthesized with a “green-chemistry” were immobilized with the CCP, which was used as a probe, on the sensing surface through 3-aminopropyltriethoxysilane and GA. The use of GA and the green-chemistry synthesis of the IONPs make this platform simple and cost-effective. The obtained results showed a linear range from 8 to 250 pg/mL, with a limit of detection of about 15 pg/mL, thereby indicating its application to quantify anti-CCP levels in the biological fluid in order to diagnose RA [197].

Since the IgM-RF and anti-CCP are considered to be the two relevant biomarkers used for RA diagnosis, Guerrero et al. developed the first dual electrochemical biosensor for the simultaneous determination of the RF and anti-CCP autoantibodies. In this work, carboxyl- and neutravidin-functionalized magnetic beads (cMBs and Neutr-MBs) and screen-printed dual electrodes (SPDEs) were used to assemble the electrochemical sandwich sensing platform. For this, Fc(IgG) was immobilized on cMBs, and a biotinylated CCP peptide was immobilized onto Neutr-MBs for the specific detection of the RF and anti-CCP, respectively. After incubation with the target, HRP-labeled detection antibodies (HRP-IgM) and HRP-IgG were further involved to identify both the RF and anti-CCP, respectively, and generate the signal using the H_2_O_2_/HQ system as an indicator. The developed platform exhibits high sensitivity for the RF and anti-CCP with LOD values of 0.8 and 2.5 IU/mL, respectively. The dual electrochemical system was also tested for the determination of both the target biomarkers for human serum. In addition to its simplicity, the biosensor can reduce the sample volume by four times when compared with that required by the ELISA method [198].

Different electrochemical biosensors have also been reported for the detection of interleukin-6 and TNF-α [199,200,201]. In 2018, Yagati et al. developed a label-free impedimetric biosensor for TNF-α detection using microdisk electrodes. For this, an ITO disk electrode was modified by electrodeposition of AuNP-decorated, reduced graphene oxide (rGO). After that, a TNF-α antibody (Ab-TNF-α) was immobilized on AuNPs/rGO/ITO through amine coupling reactions. The detection mechanism was principally based on the resistance changes of the [Fe(CN)_6_]^3−/4−^ redox probe’s movement toward the anti-TNF-α antibody/AuNP-rGO/ITO. The immunosensor possesses a linear range of 1–1000 pg/mL with a detection limit of 0.67 pg/mL and 0.78 pg/mL for PBS and human serum, respectively [202]. Given their high affinity and chemical stability, aptamers have also been used to develop electrochemical biosensors for IL-6 and TNF detection [203]. Torkzadeh-Mahani and his colleagues reported an electrochemical aptasensor for tumor necrosis factor *α* detection using an aptamer–antibody sandwich structure. For this, a graphite screen-printed electrode surface was modified by cobalt hexacyanoferrate (CoHCF) and AuNPs, which was followed by the immobilization of a TNF-α aptamer. After the binding of TNF-α, the anti-TNF-α antibodies labeled with HRP acts as a detection probe, which was added to form the sandwich-like structure. Hydrogen peroxide and the o-amino phenol were used as a probe system for HRP activity to generate the signal. The impedimetric measurements showed a linear concentration range between 1 and 100 pg/mL, with a LOD of 0.52 pg/mL [204].

Later in 2022, Yuan and his colleagues developed a miniaturized labeled electrochemical magneto-immunosensor (ECMIS) for the ultrasensitive detection of interleukin-6. For this, the micro-fabricated working electrode was electrochemically functionalized with a combination rGO and AuNPs. Then, the capture anti-IL-6 antibodies were bounded with magnetic beads to detect IL-6. The obtained magnetic beads were immobilized on the electrode working surface. After incubation with the target, a bioinylated detection antibody (dAb) was further used to bound the streptavidin-labeled HRP and generate the signal. The results showed a linear range from 0.97 to 250 pg/mL and a LOD of 0.42 pg/mL. The applicability of the developed sensing system was also demonstrated in real serum samples. The rGO and AuNPs were principally used to increase the surface area and enhanced the charge transfer rate, thus resulting in a more sensitive immunosensor [205].

### 4.4. Systemic Lupus Erythematous Sensing Strategies

SLE is a chronic inflammatory disease that is characterized by the presence of antibodies against different organs of the human body. Different biosensors have been developed for the simple and rapid diagnosis of this AD. However, in comparison with the above discussed biosensors, SLE did not present a large array of biosensors in the literature [206].

In 2017, Gimenez-Romero et al. developed the first label-free multiplex piezoelectric biosensor for monitoring the anti-TRIM21 and anti-TROVE2 circulating autoantibodies, which are common autoantibodies that target autoantigens TRIM21 and TROVE2, for SLE diagnosis. In this work, TRIM21 and TROVE2 autoantigens were covalently immobilized on a quartz crystal microbalance with dissipation monitoring (QCM-D) for the sensitive detection of their specific autoantibodies. The proposed platform exhibits a linear range of 0.32–7.17 U/mL (anti-TRIM21) and 0.07–1.46 U/mL (anti-TROVE2), with a LOD of 0.01 U/mL and 0.005 U/mL for anti-TRIM21 and anti-TROVE2, respectively. Based on the obtained results, we can conclude that the use of a multichannel quartz crystal microbalance array with dissipation as the monitoring platform for SLE diagnosis offers several advantages, including simplicity, ultrasensitivity, and a low cost [207].

In 2018, Fagundez et al. Developed a sandwich amperometric immunosensor for the detection of anti-dsDNA autoantibodies. For this, they immobilized a dsDNA on the surface of a screen-printed carbon graphite electrode. During incubation with the target, the anti-mouse IgG antibodies conjugated with a HRP enzyme were added to generate the signal. After the addition of TMB/H_2_O_2_ as an enzyme substrate, the electrochemical measurements showed a LOD of about 8 µg/mL. Further efforts should be considered to enhance the LOD and automatize the biosensor [208].

The BAFF and APRIL are also used as SLE biomarkers since they show a high serum level in patients diagnosed with SLE. Despite their positive correlation with biomarkers for anti-dsDNA antibodies, they can also predict the flare periods for patients receiving treatment [209]. For this, Pingarron and his colleagues developed a dual electrochemical immunosensor for the simultaneous detection of the BAFF and APRIL. First, the diazonium salt of 4-aminobenzoic acid (p-ABA) was electrodeposited on a SPDCE. Then, the EDC/NHS chemistry was used to covalently immobilize the biotinylated capture antibodies after the addition of neutravidin on the electrode. After incubation with the target, MWCNTs decorated with MoS_2_ were used as the nanocarrier of the detector antibodies (d-Ab) and HRP to generate and amplify the electrochemical signal (MoS_2_/MWCNTs(-HRP)-dAb) (Figure 5A). The activated MWCNTs were used to provide the functional groups required to immobilize a high number of d-Ab molecules and HRP. On the other hand, MoS_2_ was used to amplify the signal and improve the sensitivity since it presents pseudo-peroxidase activity. After the addition of the H_2_O_2_/HQ system, the amperometric measurements exhibited a linear range of 0.24–120 ng/mL for the BAFF and 0.19–25 ng/mL for the APRIL, with a LOD of 0.08 and 0.06 ng/mL for the BAFF and APRIL, respectively. Moreover, the developed dual platform is also used for the accurate quantification of both biomarkers in the serum samples of patients diagnosed with SLE [210]. In order to further improve the sensitivity of the developed dual platform, Pingarrón and his colleagues developed the first dual MB-assisted immunoplatform for the simultaneous determination of the BAFF and APRIL based on a sandwich assay [211]. In this work, biotinylated anti-BAFF and anti-APRIL capture antibodies were implemented on magnetic microparticles functionalized with neutravidin (NAMBs) or carboxylic groups (COOH-MBs), respectively. The detection of the BAFF and APRIL were performed, respectively, using a secondary detection antibody enzymatically labeled with anti-mouse immunoglobulin HRP (HRP-anti-mouse IgG) or streptavidin HRP (Strep-HRP). The resulting conjugated MB was captured by the working electrode surface (WE1 or WE2) of a SPDCE (Figure 5B). In the presence of the H_2_O_2_/HQ system, the amperometric measurements showed a wide linear range of 1.1–100 pg/mL (BAFF) and 0.05–20 ng/mL (APRIL) with lower LOD values of 0.33 and 16.4 pg/mL for the BAFF and APRIL, respectively. In addition to its simplicity and applicability, the developed MB-assisted dual immunoplatform provides higher sensitivity in comparison with that in the first report, which is only based on electrochemical platforms.

## 5. Conclusions

Nowadays, ADs present a serious health issue given the annual increase in reported cases and the absence of available cures for most of them. Therefore, the need of establishing a highly sensitive and reliable diagnostic tool for ADs is considered to be a big challenge. For this, biosensors are considered to be attractive and valuable tools for the diagnosis of ADs as they provide lower cost analysis, a fast response, and real-time detection with high sensitivity and selectivity.

Based on the studies discussed in this review, we can say that AD-based biosensors still suffer from some limits including a complex and time-consuming fabrication process, high cost, non-applicability to real samples, and the presence of large errors during measurements. Therefore, they need further improvements in terms of the simplicity of the fabrication process, multiplexed detection, cost-effectiveness, and their application for real samples. The integration of cutting-edge technologies, such as CRISPR-based biosensors and nanomaterials, holds promise for addressing current challenges and expanding the scope of biosensor applications. Moreover, as research in biosensors for AD progresses, the use of artificial neural networks, such as machine learning algorithm-based artificial intelligence, may offer a promising tool for the continuous improvement of biosensors and the development of real-time and user-friendly diagnostic tools. According to the literature we can say that there is no report on artificial intelligence-assisted biosensors for AD diagnosis. Thus, much effort is still required to establish them as an appropriate platform for the POC diagnostics of ADs.

## Figures and Tables

**Figure 1 sensors-24-01510-f001:**
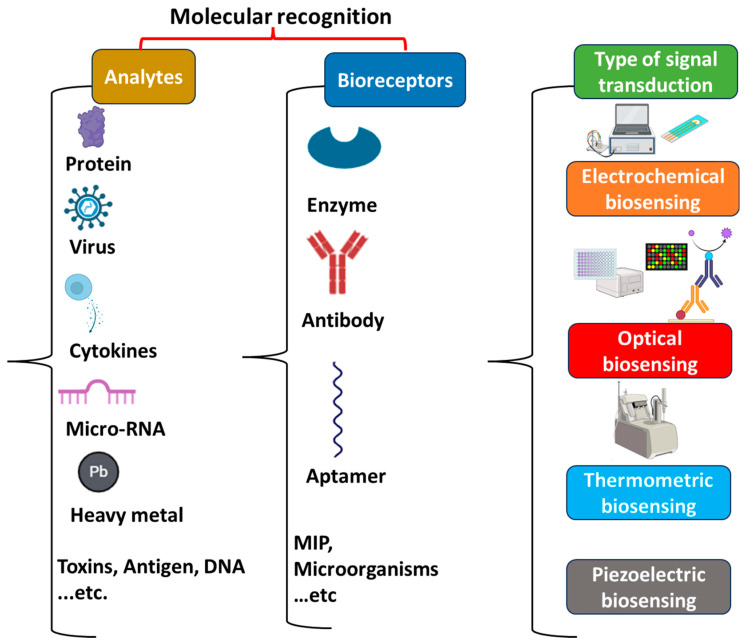
Schematic illustration of the main components of a biosensor.

**Figure 2 sensors-24-01510-f002:**
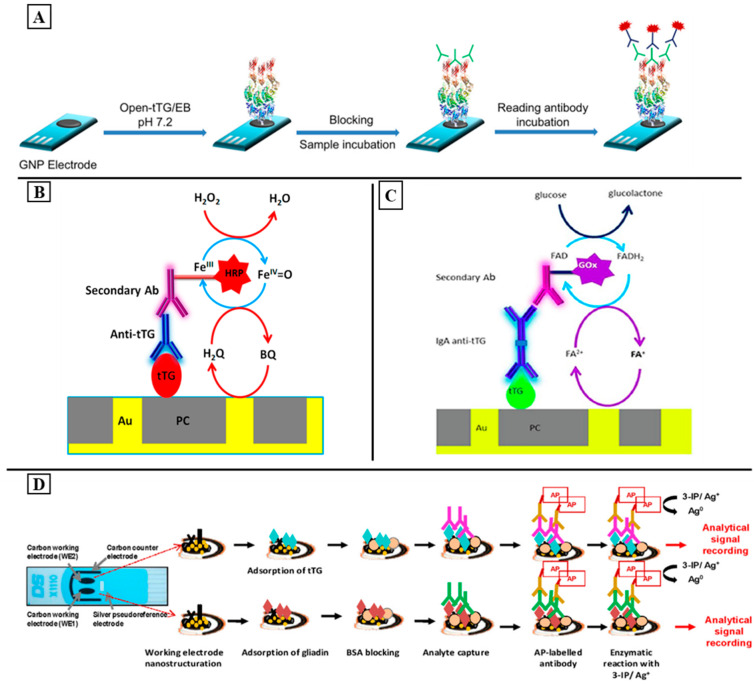
Schematic representations of the following: (**A**): A portable device exploiting the integration of screen-printed electrode-based immunosensors and remote-controlled IoT-WiFi for IgA and IgG anti-tissue transglutaminase detection. Reproduced with the permission from Elsevier (Amsterdam, The Netherlands) [173]. Copyright © 2018 Elsevier B.V. All rights reserved. (**B**): Electrochemical immunosensor based on nanoelectrode ensemble (NEE) technology for the detection of the IgG-TGA (PC = polycarbonate; tTG = tissue transglutaminase; anti-tTG = antibody for tissue transglutaminase; HRP = horseradish peroxidase; H_2_Q = hydroquinone; BQ = benzoquinone) [112]. (**C**): Novel NEE biosensor for the detection of IgA-TGA (FA^2+^ = (ferrocenylmethyl)trimethylammonium, Gox = glucose oxidase). Reproduced with permission from Elsevier, [174]. Copyright © 2021 Elsevier B.V. All rights reserved. (**D**): Electrochemical dual immunosensor for the simultaneous detection of IgA and IgG types of AGA and TGA antibodies using a carbon–metal hybrid system as the transducer surface (tTG = tissue transglutaminase; AP = alkaline phosphatase; 3-IP: 3-indoxyl phosphate; Ag^+^ silver ions). Reproduced with permission from Elsevier, [175]. Copyright © 2012 Elsevier B.V. All rights reserved.

**Figure 3 sensors-24-01510-f003:**
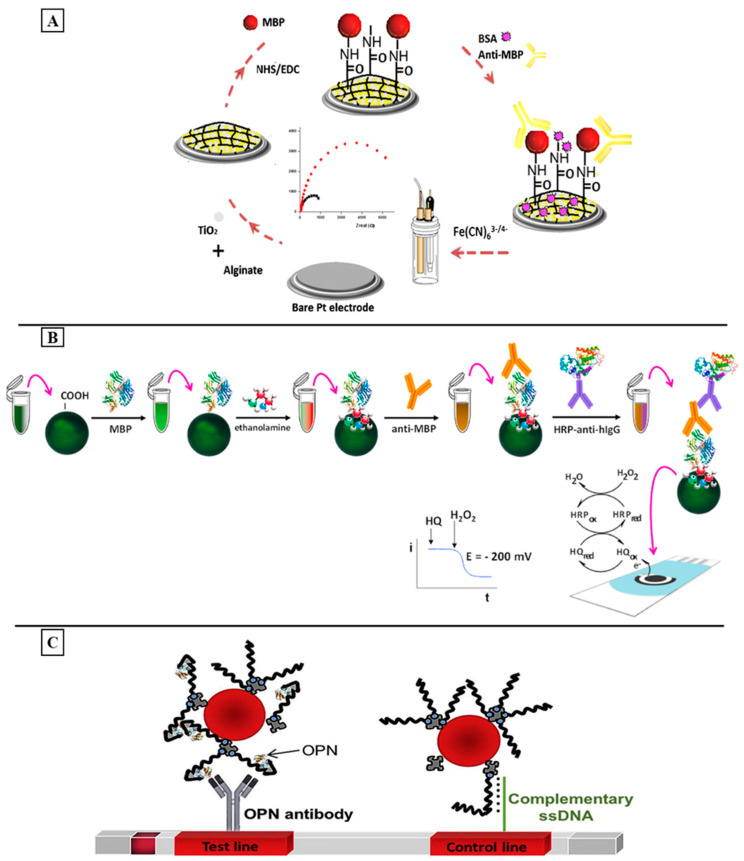
(**A**): Label-free electrochemical immunosensing platform for anti-MBP detection based on the alginate-titanium dioxide (TiO_2_) nanocomposite-film-modified platinum (Pt) electrode. Reproduced with permission from Elsevier [181]. Copyright © 2013 Elsevier B.V. All rights reserved. (**B**): Schematic representation of the different steps involved in the construction of the magnetic MBs-based immunosensor for anti-MBP detection involving MBP immobilization onto cMBs and specific conjugation with the target antibody followed by HRP-anti-hIgG conjugation and amperometric detection in the presence of the H_2_O_2_/HQ system (cMB: carboxylated magnetic microparticle; HRP = horseradish peroxidase; HQ = hydroquinone; H_2_O_2_ = hydrogen peroxide; hIgG = human immunoglobulin G). Reproduced with permission from Elsevier, [182]. Copyright © 2022 Elsevier B.V. All rights reserved. (**C**): Illustrative scheme of a lateral flow aptasensor for osteopontin (OPN) detection. The OPN-aptamer complex-containing samples were subjected to the LFB sample pad flow and reacted with AuNPs-SA on the conjugate pad; they were then subsequently captured by the anti-OPN antibody at the test line. Then, the excess biotinylated OPN aptamers were captured by the partially complementary ssDNA probes immobilized on the control line. Reproduced with permission from Elsevier [183]. Copyright © 2019 Elsevier B.V. All rights reserved.

**Figure 4 sensors-24-01510-f004:**
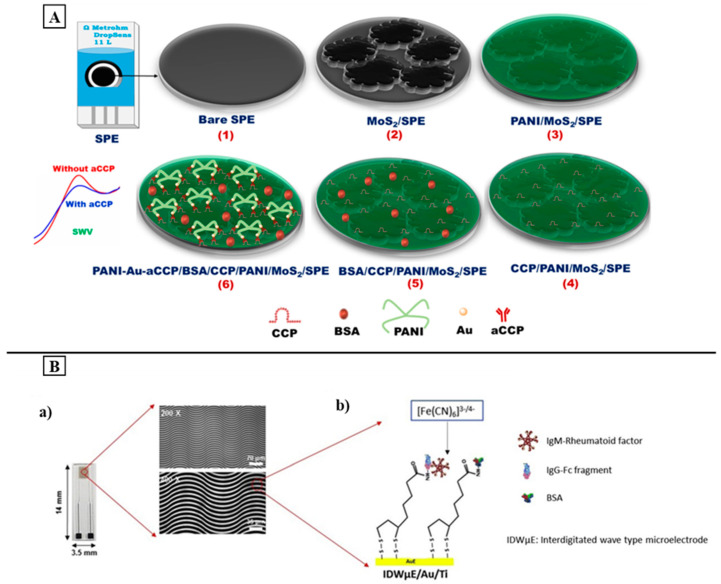
(**A**): Schematic representation of the fabrication of anti-cyclic citrullinated peptide antibodies based on polyaniline (PANI)/MoS_2_-modified screen-printed electrodes with PANI-Au nanomaterial-based signal amplification for the sensitive detection of the anti-CCP (MoS_2_ = molybdenum disulfide). Reproduced with permission from Elsevier [194]. Copyright © 2021 Elsevier B.V. All rights reserved. (**B**): (**a**) Photograph of the fabricated IDWμE with dimensions of 3.5 × 14 mm. Both low-magnification (200×) and high-magnification (400×) microscopic images of the IDWμE array show a finger width and spacing of 7 μm. (**b**) Schematic representation of SAM functionalization on IDWμE and the crosslinking of IgG-Fc fragments onto the functionalized electrode array (IDWμE = interfingered wave microelectrode array). Reproduced with permission from Elsevier [195]. Copyright © 2019 Elsevier B.V. All rights reserved.

**Figure 5 sensors-24-01510-f005:**
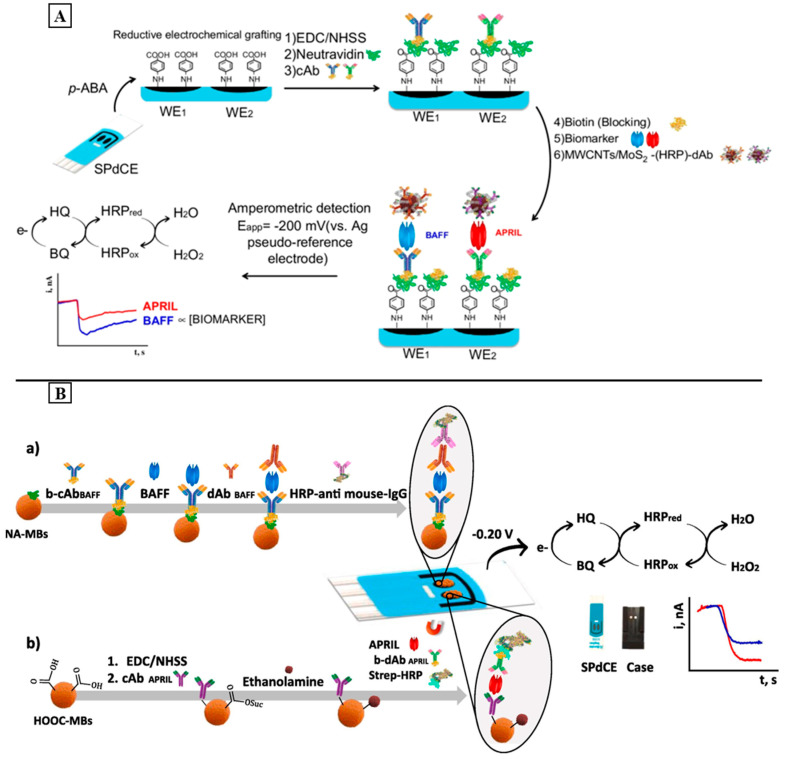
(**A**): Schematic illustration of the different steps involved in the preparation of the dual immunosensor using MWCNTs/MoS_2_(-HRP) -dAb nanocarriers for the determination of the BAFF and APRIL biomarkers by amperometric transduction (MWCNTs = multiwalled carbon nanotubes, MoS_2_ = molybdenum disulfide, HRP = horseradish peroxidase, dAb = detector antibody) [210]. (**B**): Schematic display of the different steps involved in the preparation of the developed dual immunoplatform for the (**a**) BAFF and (**b**) APRIL biomarkers as well as the reactions implied in the amperometric transduction (BAFF = B-cell activation factor, APRIL = a proliferation-induced ligand). Reproduced with permission from Elsevier, [211]. Copyright © 2022 Elsevier B.V. All rights reserved.

**Table 1 sensors-24-01510-t001:** Clinical concentration levels of some AD biomarkers.

Disease	Biomarkers	Clinical Concentration Levels	References
Celiac disease	Anti-tTG antibodies	<7.0 U/mL: normal range Between 7.0 and 9.0 U/mL: weakly positive >9.0 U/mL: positive	[112,113]
Multiple sclerosis	IgG index	0.7: the cut-off value of the IgG index >0.7: positive	[60]
	MBP	<4 ng/mL: normal range of the MBP in CSF Between 4 and 8 ng/mL: chronic myelin disruption or recovery from a relapse >9 ng/mL: myelin damage at the moment	[36]
	IL-12	0–5.0 pg/mL normal range Between 5.5 and 18.6 pg/mL: positive	[36,114]
	Osteopontin	63.74 ng/mL: normal value >63.74 ng/mL: positive	[115]
Rheumatoid arthritis	RF	0–20 IU/mL: normal range >20 IU/mL: positive	[116]
	TNF-alpha	24.47 pg/mL: normal value >24.47 pg/mL: positive	[117]
	Anti-CCP-ab	<20 IU/mL: normal value >20 IU/mL: positive	[118]

Abbreviations: TGA: transglutaminase antibody; tTG: tissue transglutaminase; IoT: internet of things; Anti-MBP: autoantibodies against the myelin basic protein; IL: interleukin; EIS: electrochemical impedance spectrometry; CSF: cerebrospinal fluid; Anti-CCP-ab: anti-cyclic citrullinated peptide antibody; RF: rheumatoid factor; TNF-α: tumor necrosis factor alpha.

## Data Availability

Data are contained within the article.

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
