# Peer review of "Recent Advances in Biosensors for Diagnosis of Autoimmune Diseases"

_sensors, 2024, doi:10.3390/s24051510_

Round 1
Reviewer 1 Report
Comments and Suggestions for Authors
In this manuscript, the authors reviewed the recent advances in biosensors for diagnosis of autoimmune diseases. Therefore, this manuscript can be published after minor revisions.
1. The principle and the different types of biosensors were discussed, which sensor was better for diagnosis of autoimmune diseases? What kind of sensors were the authors looking at? Why is that?
2. The authors overview the characteristics of biosensors based on different bioreceptors. What characteristics should the biosensors have?
Comments on the Quality of English Languagenothing
Author Response
- The principle and the different types of biosensors were discussed, which sensor was better for diagnosis of autoimmune diseases? What kind of sensors were the authors looking at? Why is that?
In our review, we discussed the principles and various types of some biosensors employed in the diagnosis of autoimmune diseases. The selection of a specific biosensor depends on several factors, including the nature of the autoimmune disease, their specific biomarker, the used materials, and the requirements for sensitivity, specificity, and time of detection. Also, we can say that electrochemical biosensors are more adapted for diseases that require the quantification of the analyte, on the other hand, we can say that optical biosensors are more suitable for the qualitative detection of certain biomarkers “presence/absence”. In this review we focused on the recent biosensors developed for AD diagnosis, that most of them are based on electrochemical detection. Also, it is noteworthy to mention that there is no ideal biosensor that encompasses only the best characteristics. Each developed biosensor has their advantages (easy to use, simple preparation, cost-effectiveness, high sensitivity…etc.) and theirs inconvenient (complex preparation, high-cost, less sensitivity…etc.).
- The authors overview the characteristics of biosensors based on different bioreceptors. What characteristics should the biosensors have?
The biosensors should have several characteristics, including: sensitivity (the bioreceptor should detect the analyte at very low concentration), selectivity (among the different existing molecules, the bioreceptor should recognize only its specific target), detection range and reproducibility. These are the principal characteristics of biosensors. Rapid response time, stability, portability, cost-effectiveness, and ease of use are also complementary characteristics that can improve the effectiveness of a developed biosensor.
Reviewer 2 Report
Comments and Suggestions for Authors
In this review, the authors summarize the recently developed biosensors for the detection of autoimmune disease biomarkers. Firstly, they focus on the main ADs biomarkers and their current method of detection. Then,the principle and the different types of biosensors are discussed. Finally, they overview the characteristics of biosensors based on different bioreceptors reported in the literature. This review topic is suitable for the scope of Sensors. Therefore, I suggest to publish this paper in Sensors. However, there are still some key comments need to be addressed.
1. The title of section 3 is suggested to be changed as “Key components of Biosensors”.
2. The authors are suggested to add a figure to introduce the content of section 3.
3. The authors are suggested to add a table to summarize and compare the methods introduced in section 4.
4. The authors are suggested to add some discussion about the emerging artificial intelligence-assisted biosensors, such as Biosensors and Bioelectronics 2022, 213, 114449; Biosensors and Bioelectronics 2023, 229, 115233.
Comments on the Quality of English LanguageMinor editing of English language required
Author Response
- The title of section 3 is suggested to be changed as “Key components of Biosensors”.
The title of section 3 is changed as “Key components of Biosensors”
- The authors are suggested to add a figure to introduce the content of section 3.
A figure to introduce the content of section 3 is added (Figure 1).
- The authors are suggested to add a table to summarize and compare the methods introduced in section 4.
A table summarizing and comparing the methods that are introduces in section 4 is added in supporting information (Table S1).
- The authors are suggested to add some discussion about the emerging artificial intelligence-assisted biosensors, such as Biosensors and Bioelectronics 2022, 213, 114449; Biosensors and Bioelectronics 2023, 229, 115233.
In the literature, there is no report about artificial intelligence-assisted biosensors for AD diagnosis. However, I mentioned this type of biosensor in the conclusion as a promising tool to enhance the analytical performance of AD based biosensors.
Reviewer 3 Report
Comments and Suggestions for Authors
A review by Marty and colleagues focuses on recent advances in biosensors for the diagnosis of autoimmune diseases. This topic is quite relevant, since autoimmune diseases are becoming increasingly widespread. This review includes many sources, however, the elaboration of the material leaves much to be desired. The introduction is organized quite well, but there are a number of questions about its main content. Here is a set of comments that must be corrected for publication approval:
1. In section 2, the authors reviewed a set of the most common autoimmune diseases and simply listed their biological markers. The organization of this section is questionable. It is not justified why only these types of diseases were selected. The authors simply listed a set of markers for their chosen disease types, without specifying what clinical concentration levels should be detected, which is critical for biosensors. Therefore, for each subsection it is worth providing a table with the clinical values of the concentrations of the described biomarkers. To ensure complete understanding, it is necessary to supplement this section with a subsection “Other diseases” and provide examples of less common autoimmune diseases with specific markers. In addition, the authors could highlight some common biomarkers for autoimmune diseases, since most of them are accompanied by inflammation.
2. Section 3 provides fairly general information about biosensors. However, this information could be organized into diagrams or illustrations to add some originality or novelty in comparison with the many similar reviews presented previously. In this section, it is also worth considering the issues of improving the key operating parameters of biosensors, especially sensitivity and specificity.
3. Section 4 also simply lists examples of work on the diagnosis of the types of autoimmune diseases chosen by the authors. However, it is not very clear why the authors used the illustrations they chose. It seems obvious that they wanted to somehow dilute the very long text. For each type of selected disease, the authors should provide a table that would summarize the latest advances in the diagnosis of the selected type of disease according to various parameters, such as transducer, bioreceptor, detection limit, detection range, etc. The text does not provide a critical review of existing biosensors for the diagnosis of autoimmune diseases, does not highlight their strengths and weaknesses, does not analyze their suitability for clinical requirements, and does not take into account possible limitations and problems of their use.
4. In the Conclusion, the authors do not conduct any analysis of the considered biosensors and do not discuss the prospects for their development and improvement of their characteristics. It is not clear from the conclusion what purpose the authors of this review pursued. One gets the impression of a rather formal attitude towards the task at hand.
Comments on the Quality of English LanguageThe Quality of English Language is fine.
Author Response
1. In section 2, the authors reviewed a set of the most common autoimmune diseases and simply listed their biological markers. The organization of this section is questionable. It is not justified why only these types of diseases were selected. The authors simply listed a set of markers for their chosen disease types, without specifying what clinical concentration levels should be detected, which is critical for biosensors. Therefore, for each subsection it is worth providing a table with the clinical values of the concentrations of the described biomarkers. To ensure complete understanding, it is necessary to supplement this section with a subsection “Other disease” and provide examples of less common autoimmune diseases with specific markers. In addition, the authors could highlight some common biomarkers for autoimmune diseases, since most of them are accompanied by inflammation.
We only selected these types of disease because they are among the most common AD [1-3]. A table of clinical concentration levels of some AD biomarkers is added (Table 1). A subsection entitled “other diseases” is also added. Some common biomarkers are cited in the section of ADs biomarkers as non-specific biomarkers such us cytokines, chemokines, the erythrocyte sedimentation rate (ESR) and the C-reactive protein (CRP).
2. Section 3 provides fairly general information about biosensors. However, this information could be organized into diagrams or illustrations to add some originality or novelty in comparison with the many similar reviews presented previously. In this section, it is also worth considering the issues of improving the key operating parameters of biosensors, especially sensitivity and specificity.
A figure is added in section 3. The sensitivity and the specificity of different biosensors are discussed in section 4.
3. Section 4 also simply lists examples of work on the diagnosis of the types of autoimmune diseases chosen by the authors. However, it is not very clear why the authors used the illustrations they chose. It seems obvious that they wanted to somehow dilute the very long text. For each type of selected disease, the authors should provide a table that would summarize the latest advances in the diagnosis of the selected type of disease according to various parameters, such as transducer, bioreceptor, detection limit, detection range, etc. The text does not provide a critical review of existing biosensors for the diagnosis of autoimmune diseases, does not highlight their strengths and weaknesses, does not analyze their suitability for clinical requirements, and does not take into account possible limitations and problems of their use.
a. Section 4 also simply lists examples of work on the diagnosis of the types of autoimmune diseases chosen by the authors
We have discussed the most recent and relevant biosensors for the detection of autoimmune disease because the aim of this review is to discuss different work of biosensors developed for autoimmune disease diagnosis.
b. However, it is not very clear why the authors used the illustrations they chose. It seems obvious that they wanted to somehow dilute the very long text
We used these mentioned illustrations because the fabrication process is complex, so to clarify the different fabrication steps and to enhance the comprehension of the used method of detection we have integrate these illustrations.
c. For each type of selected disease, the authors should provide a table that would summarize the latest advances in the diagnosis of the selected type of disease according to various parameters, such as transducer, bioreceptor, detection limit, detection range, etc.
A table summarizing the most recent biosensors with specific parameter (detection method, bioreceptor, detection limit, detection range, application in real samples) is added in supporting information (Table S1). The document is entitled Supporting Information, and it is downloaded with the main manuscript.
d. The text does not provide a critical review of existing biosensors for the diagnosis of autoimmune diseases, does not highlight their strengths and weaknesses, does not analyze their suitability for clinical requirements, and does not take into account possible limitations and problems of their use
In section 4 most of the discussed biosensors have been criticized with highlighting their strengths and weaknesses. The discussion is highlighted with a yellow color in the manuscript. For the clinical requirement, some biosensors are compared with ELISA commercial Kits and they present a more interesting analytical performance, which confirm its application. The possible limitations of some biosensors are also mentioned (example: Despite the high sensitivity and specificity obtained for the detection of TGA, no measurable response was detected when the assessments were performed in undiluted whole blood serum. In addition, this platform presents a relatively large error during measurements which limit their utilization)
4. In the Conclusion, the authors do not conduct any analysis of the considered biosensors and do not discuss the prospects for their development and improvement of their characteristics. It is not clear from the conclusion what purpose the authors of this review pursued. One gets the impression of a rather formal attitude towards the task at hand.
a. In the Conclusion, the authors do not conduct any analysis of the considered biosensors and do not discuss the prospects for their development and improvement of their characteristics
The conclusion has been modified and highlighted with a yellow color in the manuscript. Some limitations are also mentioned. The perspectives to improve the analytical performance of the AD diagnosis-based biosensors are also proposed.
b. It is not clear from the conclusion what purpose the authors of this review pursued. One gets the impression of a rather formal attitude towards the task at hand.
After modification, the conclusion becomes more informative. In summary, the goal of this review is to analyze and compare the recent developed biosensors for AD diagnosis, criticize some of them, inspire new research directions, and finally propose some interesting technologies that can enhance the performance of the future developments of biosensors.
References
- Campuzano, S., et al., Electrochemical biosensors for autoantibodies in autoimmune and cancer diseases. Analytical Methods, 2019. 11(7): p. 871-887.
- Zhang, X., et al., Immunosensors for biomarker detection in autoimmune diseases. Archivum immunologiae et therapiae experimentalis, 2017. 65: p. 111-121.
- Florea, A., et al., Electrochemical biosensors as potential diagnostic devices for autoimmune diseases. Biosensors, 2019. 9(1): p. 38.
Round 2
Reviewer 3 Report
Comments and Suggestions for Authors
The authors have done a good job of revising their review. It has become more understandable and comprehensive. In this form it can be accepted for publication.
Comments on the Quality of English LanguageThe English Language is fine.